# Ethnic Differences in Environmental Restoration: Arab and Jewish Women in Israel

**DOI:** 10.3390/ijerph182312628

**Published:** 2021-11-30

**Authors:** Diana Saadi, Izhak Schnell, Emanuel Tirosh

**Affiliations:** 1Porter School of the Environmental and Earth Sciences, The Faculty of Exact Sciences, Tel Aviv University, Tel Aviv 69978, Israel; sdianaa@gmail.com; 2Department of Geography and Human Environment, Tel Aviv University, Tel Aviv 69978, Israel; 3Bnei Zion Medical Center, (Emeritus) The Rappaport Family Faculty of Medicine, The Technion, Israel Institute of Technology, Haifa 23774, Israel; emi.tirosh@gmail.com

**Keywords:** perceived restoration scale, attention restoration theory, autonomic nervous system, cognitive response to environmental nuisances

## Abstract

Throughout the last few decades, plenty of attention has been paid to restorative environments that positively affect human psychological health. These studies show that restorative environments affect human beings emotionally, physiologically, and cognitively. Some studies focus on the cognitive effects of exposure to restorative environments. A widely used index that measures the cognitive response is the Perceived Restoration Potential Scale (PRS). Most studies employing the PRS have examined differences in human cognitive response between types of urban environments mainly urban versus green ones. We use Hartig’s questionnaire to expose differences between types of urban environments and ethnic groups. Variances between Arab and Jewish women were calculated in four environments: home; park; residential and central city environments. The effect of intervening variables such as exposure to thermal, noise, social and CO loads and social discomfort were tested. We find that dissimilar to urban typical built-up environments, green areas are highly restorative. Furthermore, differences in the restorativeness of different urban environments are low though significant. These differences depend on their function, aesthetic qualities, and amount of greenery. Ethno-national differences appear to affect the experience of restoration. While both ethnic related groups experienced a tremendous sense of restoration in parks, Jewish women enjoyed slightly higher levels of restoration mainly at home and in residential environments compared to Arab women who experienced higher sense of restorativness in central city environments. Jewish women experienced higher sense of being away and fascination. From the intervening variables, social discomfort explained 68 percent of the experience of restoration, noise explained 49 percent, thermal load explained 43 percent and ethnicity 14 percent of the variance in PRS.

## 1. Introduction

The fact that urban environments are highly stressful has received a lot of attention in recent decades. Green environments, on the other hand, have been found to have highly restorative effects on humans’ psychological and physiological indices [1,2,3,4,5,6,7].

Restoration may be defined as the ’renewal of physical and psychological adaptive resources depleted in ongoing efforts to meet the demands of everyday life’ [8]. A complementary definition relates to the psychological recovery triggered by environmental factors [9]. Two theories complement each other in explaining restorative effects: recovery from stress by psychophysiological responses and attention restoration theory leading to fatigue and cognitive failure [10,11]. Restorative environments physiologically improve the autonomic nervous system (ANS) balance [12,13,14,15], emotionally reduce stress level [16] and improve cognitive activity in daily life [17,18]. However, there is no complete isomorphism among the three aspects of physiological, emotional, and cognitive restoration, and the relationship among them remain ambiguous [19,20,21].

One of the more common indices for measuring environmental restoration is the Perceived Restoration Scale (PRS) [1,19,22,23]. These studies measured the differences in the PRS values in urban environments [1,24], with the majority pointing to the high PRS score associated with green environments compared to built-up environments. However, this assertion has not yet been fully substantiated due to limited studies that are based on small samples and that employ different methodologies and indices [25]. At the same time, not all commonly used response indices are affected in the same way by visits to green environments [25].

Furthermore, new studies highlight the restorative effect of some built-up environments [1,26,27,28,29,30], as well as small gardens in the city [31]. To allow for restoration in urban parks, a balance between enclosures that stimulate a sense of being in nature and openness that allows depth and height of sight, as well as porosity of boundaries that stimulates a sense of safety, is required [31,32,33,34]. Several studies have found that a lack of balance between closure and openness reduces the level of restoration [35,36,37].

Only a few studies have looked at how restoration experiences differ across socio-cultural groups [11]. Several studies have shown differences in ethnic groups’ physiological responses to environmental nuisances based on lifestyle, exposure to discrimination and, possibly, genetic differences [38,39,40,41]. Furthermore, minority groups may have less access to greenery for cultural or discriminatory reasons [42]. Almost all of these studies looked at emotional and physiological differences between ethnic groups in coping with environmental nuisances. We found only one exception, a study by Saadi et al. (2020). This study indicated that Arab and Jewish women cope differently with environmental nuisances as reflected by their emotional, physiological, and cognitive responses [43]. These studies support the hypotheses that argue for conceivable differences among ethnic groups’ experiences of restoration in green spaces. Since PRS appears to be one of the more common indices employed to measure the effect of greenery on human stress and health, it is important to study to what extent PRS is sensitive to ethnic differences. The fact that Arab and Jewish womens’ autonomic nervous systems respond differently to environmental challenges [43], begs the question as to whether PRS is also effected by ethnicity more relevant [11,38,39,40,41,43].

In this research, we ask the following questions: 1. To what extent do women experience deep levels of restoration as measured by the PRS in green environments compared to urban environments. 2. Do Arab and Jewish women differ in their experience of restoration in urban environments and parks? 3. Are there differences between Arabs and Jews in the relative effect of major environmental and social mediators on PRS? In the present research we aimed to elucidate the restoration effect provided by different environments on young women of two ethnic groups; Arab and Jewish and to assess the relative importance of greenery while accounting for different ambient nuisances. We further aimed at evaluating the sensitivity of the PRS in reflecting restorative effects in these two ethnic groups.

The purpose of the study is to compare groups of young healthy women of Jewish and Arab origin who live in segregated small neighboring towns in Israel in terms of their experiences with PRS in central city, residential, home and park environments. Since women and men differ in the way they respond to environmental challenges with women being more sensitive to environmental challenges compared to men [44,45], we focused our study on healthy young women only. Therefore, the main contribution of the study is in highlighting the effects of ethnicity on PRS in women in their fertile age.

### Background

Adherents of attention restoration theory argue that people are highly stimulated in urban environments, demanding high levels of directed attention. People must concentrate their efforts in order to manage a variety of complex sources of information, avoid risks, and then act effectively on the basis of such information. Long periods of concentration on a single task are likely to erode one’s ability to direct attention effectively, resulting in attention fatigue (also known as mental fatigue). The signs of directed attention fatigue include difficulty in concentrating, increased irritability, and increased rate of errors when performing tasks that require concentration [46]. The state of directed attention fatigue impairs one’s ability to effectively manage surrounding resources and daily demands, which can have devastating consequences. Furthermore, fatigue makes people more susceptible to stress, even if the initiators of stress and directed attention fatigue are different. Consequently, urban dwellers tend to accumulate mental fatigue and exhaustion, which can be released in green environments [10,47,48].

Increasing physical activity, increasing social contacts, reducing stress and paying attention to personal restoration, and reducing exposure to urban environmental stressors such as noise, air pollution, and heat load are some of the possible pathways that link restoration to green environments [49]. The greater the amount of greenery and the closer it is to the built environment in a way that does not endanger subjects, the greater the restorative effect of greenery [50,51]. However, even brief visits to small urban parks can help with physiological restoration [43,52]. Furthermore, some urban environments, such as monasteries, small gardens and green boulevards, and well-maintained city centers, may encourage some degree of restoration [10].

Personal characteristics may also influence people’s coping styles with urban environmental nuisances and restoration in green environments. Korpela and Yelen (2009) demonstrated that people of various ages and health levels experience varying degrees of restoration in green environments [53]. Pasanen et al. (2018) demonstrated that people who perceive themselves closer to nature, experience higher levels of restoration when visiting green environments [54]. Von Lindern et al. (2013) demonstrated that forest professionals experience lower levels of restoration while visiting forests [55], and [56] demonstrated that children who work in farms experience lower levels of restoration in parks [56].

Differences in coping styles with urban and restorative environments among ethnic groups may be related to lifestyle characteristics, exposure to discrimination, and, possibly, physiological differences [57]. Diet, level of activity, and clothing are all examples of lifestyle. In terms of diet, some ethnic groups follow a healthy diet consisting of whole grain meal, fish, fruits, and vegetables, while others follow a less healthy diet consisting of red meat and white grain meal [39]. Diets may also differ in terms of sugar, salt, and fat content, which may affect the risk to health [58]. Different ethnic groups may engage in varying levels of activity, which affects their metabolic rate and, as a result, their health risks [59]. Clothing insulation may have an impact on humans’ ability to deal with a thermal load [60,61]. The amount of thermal insulation a person wears has a significant impact on thermal comfort due to its effects on heat loss and, as a result, thermal balance. Layers of clothing prevent heat loss and can either help a person stay warm or cause overheating [59]. Dressing styles, on the other hand, are influenced by ethno-cultural traits that, in some cases, transcend the instinct of adjusting to the weather [61].

In terms of discrimination, several studies conducted in the United States suggest that ethnic discrimination may be associated with decreased ability to cope with stress, resulting in less favorable heart rate variability [62]. They also claim that being subjected to discrimination leads to a decrease in self-esteem and an increase in shyness, both of which increase stress and risk to health [62]. However, few (quasi) experimental studies empirically support the association between ethnic discrimination and an increase in health risks [57,63,64]. Wagner et al. (2015) studied the effects of self-reported racism on heart rate variability and confirmed Hoggard’s results that HF levels were significantly associated with exposure to ethnic discrimination, while LF levels were not.

There is some empirical evidence for ethnic related differences in physiological responses in coping with environmental nuisances. Wagner et al. (2015), on the other hand, concluded that genetic effects on coping with environmental nuisances require more evidence. Recently, Saadi et al. (2019) demonstrated some ethnic differences in the functioning of the autonomic nervous system in response to environmental challenges, presumably due to physiological differences between Arabs and Jews [42].

In addition to these differences among ethnic groups, deprived ethnic groups may have less access to green environments, either due to a lack of supply of green environments or cultural restrictions on movement to green environments. According to Saadi et al. (2019), Arab women lack access to green environments due to a lack of nearby parks as well as cultural restrictions on women’s freedom of movement [42].

Almost all of the preceding studies concentrated on the physiological and emotional effects of environments on ethnic groups’ sense of wellness and health risk. However, almost no research has been conducted to investigate the ethnic related differences of the effects of green environments on the cognitive aspects of restoration. Attention restoration theory (ART) [65] describes a potential mechanism for recovery within a restorative environment. Relaxing in an engaging environment (one that is not reliant on directed attention) is beneficial in allowing attention capacities to recover from directed attention fatigue. Restorative environments should have four characteristics in order to fulfill this restorative function: being away (from the usual environment), fascination, coherence, and compatibility [66].

In terms of the first of these characteristics, staying in restorative environments provides people with a sense of being away from daily routines that require directed attention. Forests, mountains, sea sides and meadows are all ideal places to get away in addition to historical sites, museums, etc. that may stimulate the sense of escape. Concerning the second characteristic, fascination, the mechanism of fatigue recovery involves a different type of attention, one that does not necessitate any effort on the part of the person. This ’effortless’ attention, also known as involuntary attention, is likely to resist directed attention fatigue and is central to a restorative experience. The natural environment is endowed with fascinating objects that encourage forms of exploration and keeps the person perceiving them interested without requiring active effort on their part. The other two characteristics are coherence, which represents a rich and understandable environment that stands on its own, and compatibility, which represents a person’s ability to immerse in their surroundings. They work together to create space for further exploration, assisting a person’s actions to match what the environment requires and can provide. Even brief encounters of this kind appear to have a restorative effect. Nature forms that can provide restorative benefits to an individual do not have to be more than a few trees or some indication of vegetation. Outdoor benches, for example, in a shady location, can encourage people to take a peaceful break. Outdoor sites with places to walk or opportunities to observe flora or wildlife may also present meaningful restorative benefits.

The studies that tested a variety of urban and natural environments concluded that park restoration is proportional to the amount of greenery and green space as well as the degree of closure provided by built environments without undermining the sense of security in green areas [50,51]. Aside from these criteria, studies show that historical sites, museums, and restaurants, among other places, can be experienced as restorative environments [10]. However, more research on the restorative effects of green environments on human cognitive functioning is still needed. [50,51] We hypothesize that 1. Parks are much more restorative compared to urban environments regardless of the level of activity in them; 2. Ethnic groups may differ in their level of restoration in different environments; and 3. Ethnic minorities may be more affected by social environmental factors and less by physical environmental factors as compared to participants of the ethnic majority.

## 2. Materials and Methods

### 2.1. Study Population and Location

The study’s population consisted of 72 young women; 48 were Arab women who were the core population for the original study and 24 were Jewish women who were also participating in the original study as a comparison group. They were mothers between the ages of 20 and 35 with similar BMI and middle-class affiliation. They were selected using a convenient sample from two small mono-ethnic towns in northern Israel. All were in good health, did not smoke, and did not take any medications or drugs on a regular basis.

The study area comprised 2 small towns, with a population of approximately 70,000 inhabitants each—the Arab town of Nazareth and the Jewish town of Afula. The towns are about 12 km apart in Israel’s northern peripheral region, in a Mediterranean climate (Figure 1). The towns are also inhabited by mono-ethnic groups to avoid a confounding inter-ethnic stresses in the study area. Testing density, morphology, and greenery in the studied sites in both cities based on photos, show no significant changes in the main characteristics of the environment between 2016, the year of the original study and 2021.

### 2.2. Instruments and Measurements

#### 2.2.1. Independent Variables

Noise measurements were collected using a Quest Pro-DL dosimeter. They ranged from 40 to 110 dB, and their resolution was 0.1 dB. The data were transformed to 110 dB in the few cases where noise levels exceeded 110 dB. During the 35-min visit to each environment, measurements of average noise were taken every minute. The device was worn by the researcher who followed the participants. The data were collected in a systematic manner, saved in a data logger, and transferred to a laptop. For each environment, mean measurements were computed.

The heat index was calculated based on temperature, relative humidity and radiation temperature and wind velocity and direction measurements. The temperature and relative humidity were measured using a Fourier Microlog and Kestrel 3000 devices. Radiant temperature measurements were obtained from the Israeli Meteorological Service and adjusted to account for the amount of shade in each environment. The researcher calculated the mean physiological equivalent temperature index [67] for each environment. Carbon monoxide (CO) was measured using a CO sensor attached to a Dragger Pac III, a portable device that detects changes in electrical potential during oxidation. The data were collected in a data logger from which the mean results were calculated for each person in each environment.

The accuracy of the devices was determined using the calibration methods provided by the manufacturers and through comparisons with results from the Ministry for the Protection of the Environment’s permanent station [52].

#### 2.2.2. Dependent Variable

The Potential Restorative Scale (PRS), as proposed by Hartig et al. (1997) in his revised version, was assessed using a perceived restoration scale questionnaire, which contains 26 statements and measures the subject’s experience of the environment’s restorative qualities. The 26 statements were rated on a 7-point Likert scale of agreement ranging from 0 (not at all) to 6 (extremely). The psychometric properties of this scale have been previously reported [68]. The statements were grouped into four subscales: 1. being away is measured with five items (e.g., “Spending time here gives me a break from my day-to-day routine”), 2. Fascination is measured with eight items (e.g., “I want to get to know this place better”), 3. Coherence is measured with four items (e.g., “It is a confusing place”). The remaining items measured compatibility (e.g., “Being here suits my personality”). A total average per question was calculated for each subscale and for the total perceived restoration scale (PRS). The mean score for each subscale and the final mean score were calculated.

The PRS and the social mediators’ inventory were administered to the participants in each station. They were sitting on a bench or on the grass and filed the questionnaires during their stay in the station.

#### 2.2.3. Mediating Variables 

Environmental and sociodemographic indices were used as mediating variables. The environmental measurements (noise, temperature, humidity, and carbon monoxide) were collected in situ, where the participant stayed during the tests. The sociodemographic characteristics were derived from a pre-experiment questionnaire [43].

### 2.3. Procedure

The field experiment was divided into 12 sessions, with 6 participants each. Both the Arab and Jewish groups went to Nazareth, an Arab town, and Afula, a Jewish town. The women visited three ordinary outdoor environments in each town (Figure 2, Figure 3 and Figure 4): (1) the busy town center. The pictures were taken during quiet hours but the tests were performed when the center was highly crowded. In Nazareth transportation is moving slowly in the rush hour and drivers intensively used horns. In Afula the traffic was intensive but with no blocks; in both cities the houses were populated by retail stores on the first floor and residential uses on the upper floors; (2) a quiet residential neighborhood of 3–4 stories with about 10% greenery in Nazareth and 6–8 stories with approximately 20% of space covered by greenery in Afula, in both neighborhoods the streets were quiet as the transportation was mainly local; (3) the towns’ main parks, both similar in size and vegetation cover. Schnell and Saadi reported that the parks were structurally similar, with a similar balance of closure and openness. The dominant installation in the center of the parks were playgrounds for young children and in between the grass and the trees were surrounded by pedestrian roads and benches (2014). Prior to the outdoors procedure the women were monitored in their homes with their children but not with their husbands. The town centers of Nazareth and Afula are both crowded, with Nazareth’s center being noisier and less green. The residential streets chosen in both towns were quiet. The parks were roughly the same size and were covered in trees and other greenery that provided natural shade.

To eliminate the effect of the order of sites in the field experiment, the order of visits was randomly assigned into 12 sessions of 6 women each (Figure 5). The sessions took place between January 2015 and February 2016. The sessions began at 11:00, allowing for a 1-h familiarization practice with the devices. The trials were held between 12:00 and 18:30 and consisted of visiting 4 intra-ethnic and 3 inter-ethnic sites 35 min each.

For each session, the same protocol was followed. The measurement devices were calibrated and tested for accuracy prior to each session, in accordance with the manufacturer’s instructions (see [52]). The researcher visited the participants at their homes the day before each session, walked them through the informed consent form, completed an initial socio-demographic questionnaire, explained the research protocol, and demonstrated how the devices worked. On the day of the session, around 11:00 a.m., the participants operated the measurement devices on their own and completed the questionnaire (see instruments). The participants were transported to their first station by the researcher. The researcher moved between the six participants in the group to ensure that the devices were operational and properly placed, and that the questionnaires were completed in accordance with the instructions and protocol. The questionnaire-based tests were administered by the researcher first at each woman’s home and then again at each of the 6 outdoor stations while sitting 10–20 m apart in the shade. The memory tests were also individually administered during this interval. Environmental measurements were taken in each outdoor site using three devices worn by the researcher accompanying the women. (See instruments).

Prior to each site visit the women remained in an air-conditioned car (22 °C) for a 15-min ‘washout’ period. The researcher provided assistance as needed and documented any unusual events that could have an impact on the quality of the data collected. 

The Tel Aviv University Ethics Committee in Israel granted ethical approval for this study. Before the experiment began, each participant provided informed consent by signing the appropriate forms at her home. These forms provided a thorough explanation of the research’s goals and objectives, as well as the experiment’s procedure. The researcher explained all of the measured indices in the participant’s native language (Arabic for the Arab women and Hebrew for the Jewish women). The questionnaires below were filled out by participants in their native language.

### 2.4. Statistical Analyses

In total, 504 PRS measurements were taken (72 women × 7 ethnic environments  ×  1 measurement per environment). To evaluate grouping level by women and the applicability of a multilevel model, the study evaluated the intraclass correlation coefficient (ICC). The ICC was greater than 0 (ICC  =  0.18 and 0.053 for Arab and Jewish women, respectively). Even though the magnitude of the ICC is modest, we determined that multilevel modeling was the most appropriate method to adopt. The variance subscale for the Level 2 intercept was significant.

The analysis began by calculating the effects of the environment and affiliation on the PRS. Following that, a stratified model was used to focus on differences in modes of dealing with environmental challenges based on ethnic affiliation (Arab and Jewish women). We calculated ANOVA between the affiliation groups and environments on the one hand and PRS on the other.

When the mediation analysis assumptions are met, the mediation proportion presented is the proportion of the change in mean PRS attributable to elevated levels of potential environmental nuisances (i.e., thermal load, noise level, and sociodemographic characteristics) [69]. Mediation analysis was performed to calculate the percentage of the association between environments (park, town center, and residential environment) and PRS, as explained by each of the mediators.

## 3. Results

Before we present the results, we present the reliability of the partial and overall indices of the PRS. The reliability of the perceived restoration scale in our case study is presented in Table 1. Each of the subscales of the test is highly reliable, with alpha Cronbach values between 0.90 and 0.93. This is compared to the internal reliability alpha of 0.79 that has been previously reported [68].

The PRS total and subscales means as related to ethnic and environmental factors are presented in Table 2 and Figure 6 and the ANOVA F values by environmental and ethnic factors are presented in Table 3. In line with the first hypothesis, highly significant differences among environments were recorded. Extreme differences were recorded, in particular, between parks and urban built environments. In the total sample, mean PRS values in built outdoor environments reached levels of approximately one or less, and in parks, levels of 5.6 were recorded (a scale from 0 to 6). In contrast, the difference in PRS values between busy city centers and quiet residential areas was less than 1 (0.6 and 1.1, respectively). Low, albeit significant differences in the PRS results, especially in coherence and compatibility in the park restorativeness, were found. The PRS scale also distinguished between experience at home and at the park, identifying the park as the most restorative environment both for Jews and Arabs (Table 2).

In respect to the second hypothesis, it appears that Jewish women enjoy slightly higher levels of PRS mean scores, mainly at home and at residential environments, while Arab women feel stronger level of restorativeness in central city environments. They also differ in PRS subscales scores. While Jewish women experience stronger sense of being away and fascination Arab women experience stronger sense of coherence. (Table 2 and Figure 4).

Table 3 shows that the differences between Arabs’ and Jews’ sense of restorativeness are significant for the total mean PRS and “being away” subscale only. However, the interaction between environment and ethnicity explained between 17–53 percent of the variability (Table 3).

Post-Hoc environmental factor: all differences are significant at level of 0.0001 except for intra- and inter-ethnic affiliation in residential environments.

The differences between the environments were highly significant, owing primarily to the most pronounced difference between mean total PRS in parks and the rest of the environments (significance levels of 0.0001 in all subscales of PRS) (Table 4).The comparison of the ethnic related restorative experience in the alien environments between the two groups reveals significant differences (Table 4). Both Jewish and Arab women experience about the same reduction in PRS score while reaching out to alien inter-ethnic environments. Among Arab women, the differences are much more pronounced.

In testing the effect of the mediating factors on the distribution of PRS scores for all environments in line with the third hypothesis, we applied multiple-regression analyses. The independent variables were physical aspects of the environment: exposure to thermal, noise and carbon monoxide loads; exposure to social environmental variables: social discomfort, sociodemographic status, status in the family, freedom of movement to parks and access to gardens. Four variables had a significant effect on PRS in the following order: social discomfort, noise, thermal load and last entered was ethnic affiliation (Table 5).

Table 6 compares the effect of the mediating variables on Arab and Jewish women’s PRS scores. Both regressions presented strong correlation coefficients between the mediators and PRS scores (R^2^ = 0.84 and 0.86 for Arabs and Jews respectively). Four variables affected PRS scores in both ethnic groups. However, among Arabs, social discomfort and noise were the stronger mediators, while among Jews, thermal load was the main mediator.

## 4. Discussion

The sense of restoration as experienced in urban environments has been previously investigated [3,4,5,6,12,70,71]. Our study, addressing the intricate effect of ethnicity and environmental factors on the perceived restoration builds on previous research using PRS indices and adds to the discussion by focusing on ethnic differences in experiencing restoration in urban environments. The methodology employed in the present investigation ascertained the similarity of both residential and park environments in the intra- and inter- ethnic sites. Parks in Jewish and Arab towns are similar in size, greenery and shade, facilities, and maintenance levels—factors that may influence park restoration [50,53,72]. Therefore, the effect of social factors may be uniquely identified in analyzing ethnic differences in response to the studied environments.

In accordance with our first hypothesis, we demonstrate that urban parks are highly effective environments for sense of restoration. Several studies have broadly confirmed this conclusion [73,74]. Furthermore, we argue that built environments may have perceived restorative effects on humans [75]. Our findings suggest that ordinary urban environments—residential and city centers—promote very low levels of restoration. This is consistent with previous literature [24,71] It does not, however, rule out the possibility of specific urban built environments gaining restorative power. demonstrated that restoration in urban environments is directly associated with the quality of architecture and design of the environment [75]. Furthermore, particular urban environments may exhibit high restorative effects. For example, panoramic, historical, and recreational places were found to stimulate a sense of restoration [76,77]. Similarly, museums and churches appear to contribute to a sense of restoration [65]. This discussion’s main conclusion is that the restorativeness of urban environments is determined by their function, aesthetic qualities, and amount of greenery.

According to our second hypothesis, we find significant, albeit minor, differences in Jewish and Arab women’s sense of restoration in urban and green environments. Although it appears that to some extent that Jewish women feel stronger sense of restoration compared to Arab women, the interaction between ethnicity and restoration was most notable as related to the PRS subscales of fascination compatibility and coherence. This is not surprising given that social factors from which minorities tend to suffer more, appear to have a strong influence on human stress and health risks [78,79]. Our results suggest that while visiting parks Jewish women experience more fascination whereas Arab women enjoy a greater sense of being away. It is conceivable that Parks, for Arab women are first and foremost an “escape” from the parental stress they experience at home [42]. We previously reported that because they arrive at the park in groups, oftentimes with their children, they do not have time to be captivated by the park’s natural scenery [80].

In terms of PRS, Arab and Jewish women face similar feelings in crossing ethnic boundaries. They do, however, differ in their perceptions of the various subscales of PRS. Arab women lose their restorativeness in all aspects in Jewish parks, whereas Jewish women lose their sense of coherence and compatibility in Arab parks. Schnell and Saadi (2014) exposed the strong effect of visiting Jewish parks on Arab compatibility [80]. The findings of this aforementioned report suggest that Arabs visit Jewish parks because there are few parks in Arab towns. They testify that they come in large groups and demonstrate their Arab identity by listening to loud Arab music and behaving extrovertly. As a result, they signify the park with their identity as Arabs. As a result, Jews are offended by such behaviors and develop negative attitudes towards the Arabs’ presence in the parks. Negative reactions to minorities visiting majority parks have been documented in other cases [73,81,82,83,84,85,86].

In accordance with the third hypothesis, Arab and Jewish women also differ in their sensitivity to the mediating variables. In general, two physical factors, noise, and thermal load, and two social factors, sense of social discomfort and ethnic affiliation, accounted for more than 70% of the variability in PRS. While Arab women were more sensitive to social discomfort and noise, Jewish women were more susceptible to thermal load. Arab women’s sensitivity to noise is understandable since Arab homes and outdoor environments in Nazareth (the studied Arab town) reached average values of 73 dB. The average noise level at home and in the city center reached 90 dB. In comparison, the average noise level in Afula (the Jewish studied town) reached 63 dB. The average levels of noise at home reached 49 dB, and in the city center, noise of 71 dB was documented. With a standard threshold of 65 dB, Arab women are exposed to stressful noises in all visited environments, including homes, whereas Jewish women are mostly exposed to noise levels that are below threshold.

Arabs’ sensitivity to social discomfort may reflect their perception as a disadvantaged minority in Israel. A similar trend has been identified in the effects of social discomfort on HRV in studies performed in the US and Israel [38,63].

Although all PRS scores in built-up environments are low, they are even lower in Arab environments. These differences in PRS scores in built-up environments possibly reflect the chaotic structure and the lower maintenance of Arab towns in Israel when compared to Jewish towns. One of the participants commented “In neighboring Jewish towns, even the cemeteries look nice with gardens, unlike our chaotic towns”. However, a more detailed study that focuses on the environmental factors that affect restoration is required in future studies.

### Limitations

The study has the following limitations. While gender and age may influence human responses to environments, our study focuses solely on young females. More research is needed to test a broader range of ages, as well as males. The study could also benefit from testing the restoration effect of various other urban built-up environments. Similarly, it is advisable to test the differential restorative effect of parks with different structures.

## 5. Conclusions

This study confirms that green environments are restorative, whereas ordinary urban environments, with the exception of a few unique built-up sites, are much less so. It appears that the restoration provided by greenery, as experienced by healthy women and reflected by a reliable questionnaire, transcends ethnicity. It is worth noting that the cognitive processes that contribute to perceived restoration differ significantly when ethnic boundaries are crossed and different environmental factors are present. Arabs are more affected by factors such as noise and discrimination, whereas Jews are primarily affected by thermal load. The strong effect of noise on Arab women is explained by the Arab environments being much noisier than Jewish environments. At the same time, Arab women as a deprived minority, are more sensitive to crossing ethnic boundaries. Furthermore, it appears that the park restorative effect is ethnic specific as Arab women response is attributed mainly to the experience of “being away”, from the stresses at home whereas in Jewish women the restorative response is attributed mainly to fascination from the park environment.

## Figures and Tables

**Figure 1 ijerph-18-12628-f001:**
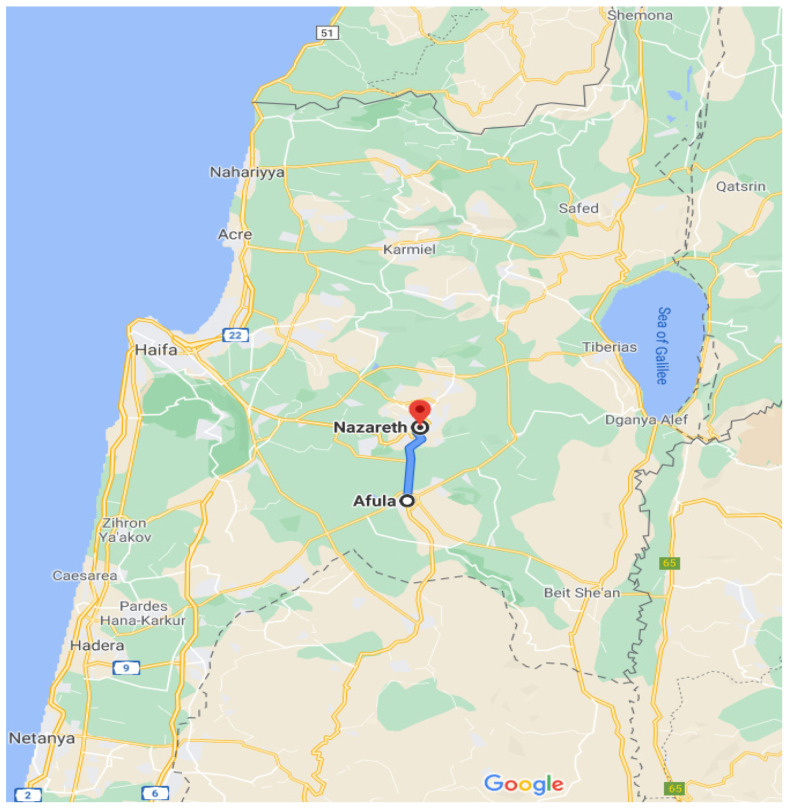
The cities locations.

**Figure 2 ijerph-18-12628-f002:**
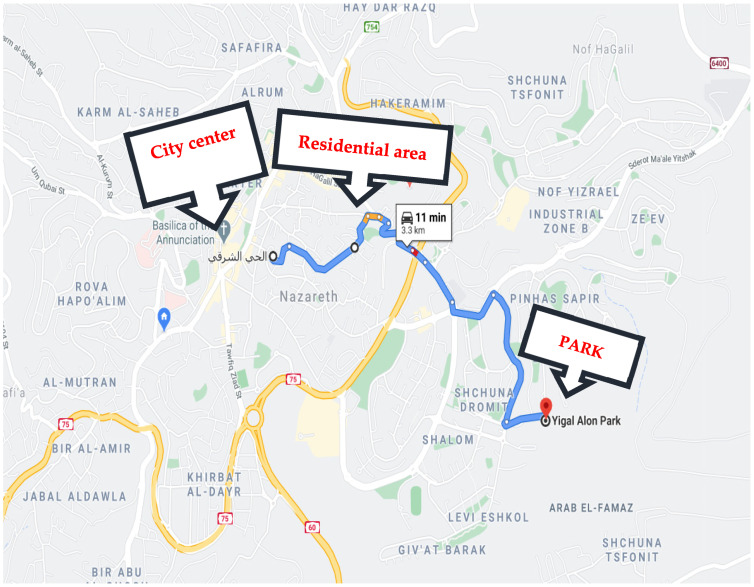
Study sites Nazareth.

**Figure 3 ijerph-18-12628-f003:**
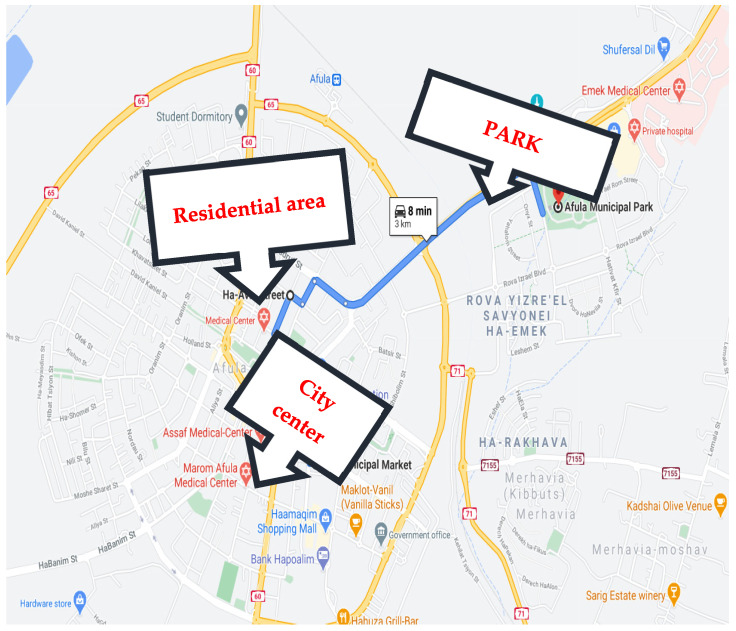
Study sites Afula.

**Figure 4 ijerph-18-12628-f004:**
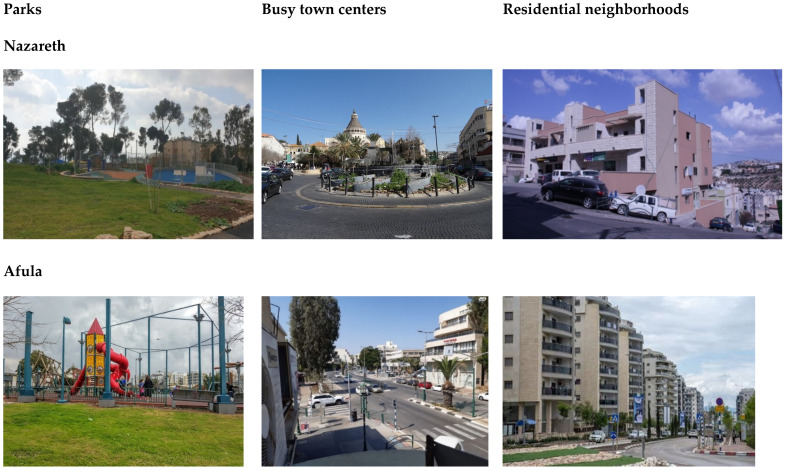
The three environments in Nazareth and Afula.

**Figure 5 ijerph-18-12628-f005:**
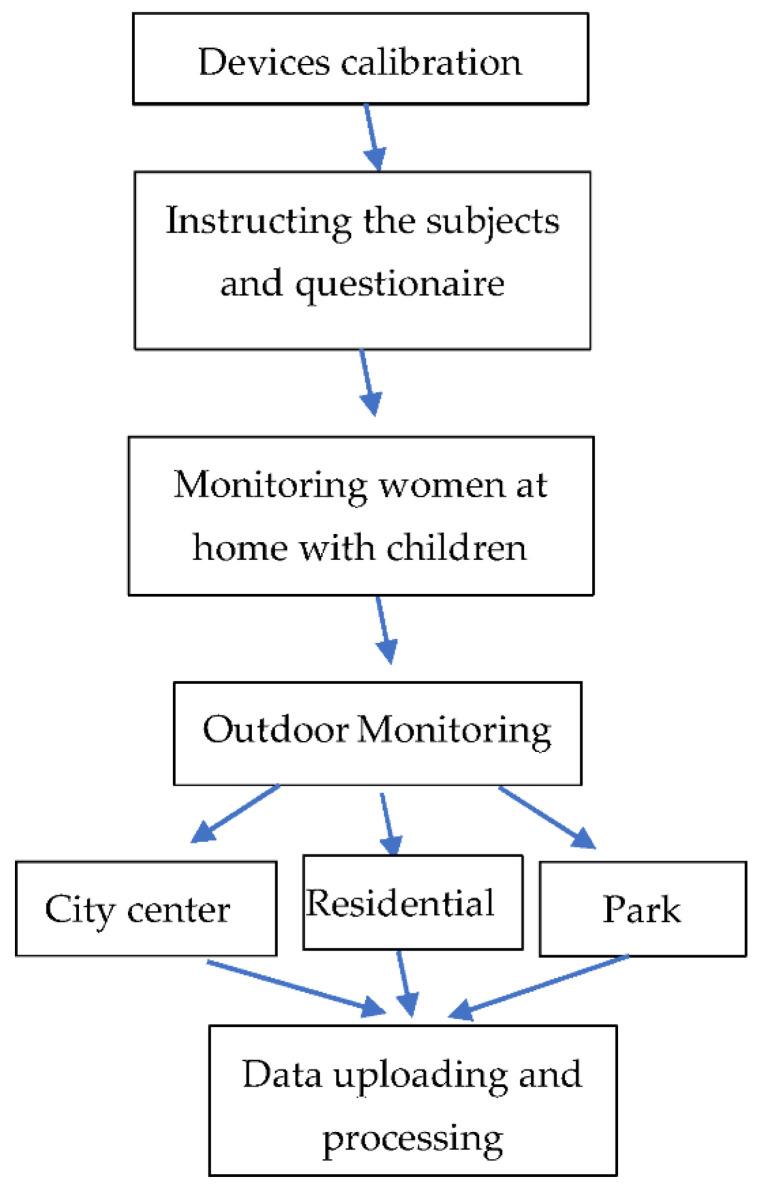
The stages of the study.

**Figure 6 ijerph-18-12628-f006:**
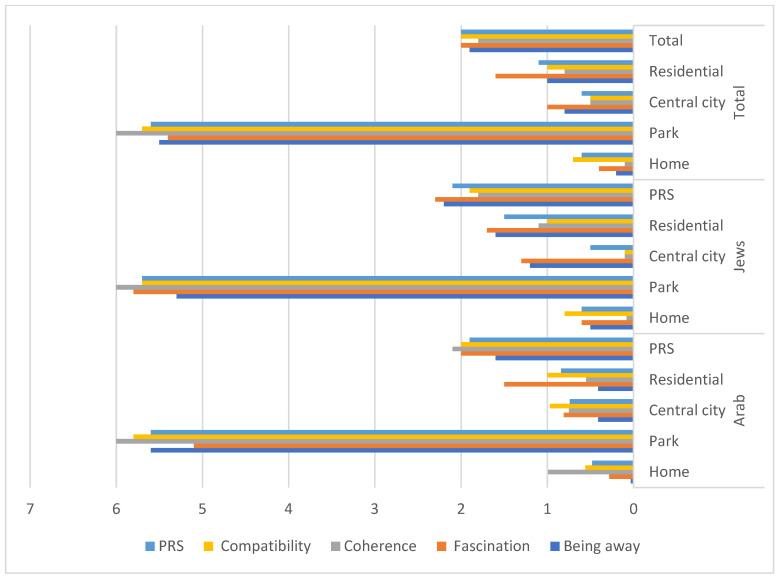
Mean restoration values for PRS indices by ethnic and environmental factors.

**Table 1 ijerph-18-12628-t001:** Cronbach Alpha Reliability Coefficient.

	Being Away	Fascination	Coherence	Compatibility	PRS
Cronbach α	0.92	0.90	0.90	0.92	0.93
No. of Items	5	8	4	9	26

**Table 2 ijerph-18-12628-t002:** Mean restoration values for PRS indices by ethnic and environmental factors.

Ethnicity	Environment	PRS Aspects				
Arab		Being away	Fascination	Coherence	Compatibility	PRS
	Home	0.03	0.28	0.99	0.56	0.48
	Park	5.6	5.1	6.0	5.8	5.6
	Central city	0.41	0.81	0.75	0.97	0.74
	Residential	0.41	1.5	0.55	1.0	0.84
	Total	1.6	2.0	2.1	2.0	1.9
Jews	Home	0.5	0.6	0.08	0.8	0.6
	Park	5.3	5.8	6.0	5.7	5.7
	Central city	1.2	1.3	0.1	0.1	0.5
	Residential	1.6	1.7	1.1	1.0	1.5
	Total	2.2	2.3	1.8	1.9	2.1
Total	Home	0.2	0.4	0.1	0.7	0.6
	Park	5.5	5.4	6.0	5.7	5.6
	Central city	0.8	1.0	0.5	0.5	0.6
	Residential	1.0	1.6	0.8	1.0	1.1
	Total	1.9	2.0	1.8	2.0	2.0

PRS: Perceived Restoration Scale.

**Table 3 ijerph-18-12628-t003:** ANOVA of ethnic and environmental differences by index of PRS.

Index	Statistics	Effects		
		Environmental (df = 6)	Ethnic (df = 1)	Env *Ethnic
PRS	F	8111	25.9	39.3
	Sig.	0.0001	0.0001	0.0001
	Partial ETA^2^	0.90	0.05	0.32
Fascination	F	6727	1.6	33.2
	Sig.	0.0001	0.2	0.0001
	Partial ETA^2^	0.90	0.003	0.29
Being away	F	3185	89.8	94.1
	Sig.	0.0001	0.0001	0.0001
	Partial ETA^2^	0.90	0.16	0.53
Compatibility	F	6728	1.6	33.2
	Sig.	0.0001	0.2	0.0001
	Partial ETA^2^	0.92	0.003	0.29
Coherence	F	883	0.7	17.1
	Sig.	0.0001	0.4	0.0001
	Partial ETA^2^	0.89	0.001	0.17

**Table 4 ijerph-18-12628-t004:** Restoration in parks by ethnic affiliation, PRS subscales and crossing ethnic boundaries.

Restoration in Parks	Jews				Arabs			
	Mean		Difference		Mean		Difference	
	Inter-Ethnic	Intra-Ethnic	F	Sig.	Inter-Ethnic	Intra-Ethnic	F	Sig.
PRS	5.0	5.7	7.6	0.001	4.8	5.5	15.4	0.0001
Fascination	5.5	5.8	6.4	0.002	5.1	5.1	18.3	0.0001
Being away	5.1	5.5	6.4	0.002	4.6	5.9	5.1	0.001
Compatibility	4.8	5.7	8.6	0.0001	4.4	5.8	18.2	0.0001
Coherence	3.9	5.8	11.5	0.0001	4.4	5.9	6.2	0.002

PRS: Perceived Restoration Scale.

**Table 5 ijerph-18-12628-t005:** The effects of mediating variables on PRS.

		Beta	*t*-Test	Sig.	Partial Corr.
Physical environ.	Thermal load	−0.34	−10.6	0.0001	−0.43
	Noise	−0.44	−12.7	0.0001	−0.49
	CO	0.26	0.85	0.39	0.04
	Garden at home				Excluded
	Park in town				Excluded
Affiliation	Arab/Jew	−0.28	−3.2	0.002	−0.14
Sociodemographic	No. children	−0.04	−1.3	0.18	
	Work				Excluded
	Economic difficulties	−0.03	−0.	0.9	−0.3
Status in family	Decision making	0.03	−0.6	0.56	0.3
	Freedom of movement	0.03	0.4	0.7	−0.2
	Social discomfort	−0.38	−13.9	0.0001	−0.68

R^2^ = 0.72; F = 141; sig. = 0.0001.

**Table 6 ijerph-18-12628-t006:** Mediators’ effects on restoration.

Mediators Effect	Arab Women		Jewish Women	
	Beta	Sig.	Beta	Sig.
Social discomfort	−0.5	0.0001	−0.09	0.006
Thermal load	−0.13	0.0001	−0.87	0.0001
Noise	−0.51	0.0001	−0.23	0.0001
CO	+0.23	0.0001	+0.27	0.0001

R^2^_Arabs_ = 0.84; F = 219; Sig. = 0.0001; R^2^_Jews_ = 0.86; F = 202; Sig. = 0.0001.

## Data Availability

Data can be available by direct request from the corresponding author.

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
