# Peer review of "Ethnic Differences in Environmental Restoration: Arab and Jewish Women in Israel"

_ijerph, 2021, doi:10.3390/ijerph182312628_

Round 1
Reviewer 1 Report
This version can be accepted for publication.
Author Response
No changes requested

Reviewer 2 Report
Dear colleagues,
The paper has been improved, but not so much as to convince me to publish it in this form.
New citations have been included in the introductory chapter, but they are not quoted correctly in the text, according to the magazine's standards. Here are just a few examples: line 67 (Saadi et al. 2019), line 70 (Saadi et al. 2020), line 77 (Saadi et al. 2019), line 93 (Woo, 2015; Shafer, 93 et ​​al. 2017). ), line 119 (Weber and Trojan, 2018) etc. Quotations in the text are made with numbers.
In terms of methodology, you still haven't convinced me when you did the field research. If in the previous version the year 2016 was mentioned, in this new version no exact date is mentioned regarding this aspect.
This phrase does not seem at all convincing to me: Testing crowding, morphology, greenery in the studied sites in both cities show no significant changes in the main characteristics of the environmentbetween 2016-2021. I still tend to think that you kept only the data collected from 2016. These aspects are related to professional ethics, nothing more.
This figure (Figure ?: The three environments in Nazareth and Afula) has no number and no source with whom he took the photos ?! Then the next unnumbered figure (Figure ?: stages of the study) ?!
In the end, I would have liked more professionalism from the authors.
Sincerely,
Author Response
Reviewr 2:
- New references incorporated according to Journal style. For now all references conform the journal style.
- Line 220 we mention the year of the original study.
- We corrected reference 42.
- Apology for the lack of clearance. All the data are from 2016 and we hoped it was clear from the text. We added a sentence based on analysis of photos to argue that the environmental characteristics did not change since then. We believe that to present these photos is out of the main argument for such an article.
- We added numbers to all figures and maps.
Reviewer 3 Report
The article is very interesting. Very thorough statistical analysis. However, I have the following question.
Changing what landscape element would have the most beneficial effect on restorative in urban areas.
Why the study was not conducted in different age groups?
Would conducting a study for another religion change the results strongly?
Could a change of country strongly affect the results?
Please describe the locations analyzed in more detail in the text
Author Response
Reviewer 3:
- So far we do not have any systematic analysis of associations between landscape factors and restoration. We added at the end of the discussion section a sentence calling for further studies in this direction.
2-5. This is one of the first studies to search for differences among social groups in experiencing restoration. We cannot predict if other ethnicities, places, country or gender will make differences. From this study we learn that there is support for hypothizing in this direction.
5 We tried to add some details about the study environments on lines 274-283.
This manuscript is a resubmission of an earlier submission. The following is a list of the peer review reports and author responses from that submission.
Round 1
Reviewer 1 Report
I have the following concerns for the research:
(1) Why do you conduct this study?
(2) What are your contributions?
(3) Why do you choose young healthy women of Jewish 89 and Arab
(4) The section of theoretical review seems not to involve in any theories. If you want to present a theoretical review or framework, please construct a theoretical consideration or system to guide your study and empirical section. If you just want to do a literature review, you just state why do you conduct the literature review and how do you expand your literature review.
(5) You should justify why you choose the population and location.
(6) How do you conduct or develop the a stratified model in your research? Please offer some model steps with respect your study.
(7) Because you have no clarified steps of model, I cannot understand the section of results. I can't understand where the results reflect the stratified model.
(8) In discussion section, you mention “first hypothesis”, so where and what are your hypothesis?
(9) You should write your discussion by linking to your results.
Reviewer 2 Report
Dear colleagues,
The study is substantiated, the results are conclusive, but I have some doubts:
Line 84: “In this context, we ask the following questions”: I suggest that it would be more appropriate to use the term “research” next to questions.
I understood that 72 young women participated in the study. Why 48 Arab women and only 24 Jewish women? How do you argue this? What was the selection principle? I suggest you go into more detail so that the reader understands the whole context.
Also, the methodology shows that the field data were collected between January 2015 and February 2016. Are these data still one hundred percent valid today? Has nothing of what you analyzed changed within the two communities (Arab and Jewish) changed? I think some newer data should be collected from the field, it's been six years since then.
Sincerely,
Reviewer 3 Report
The content of the article is interesting. The research required a lot of time. However, the content of the paper needs improvement. I have provided comments and feedback on the article below.
Misquoted lietrature in the text of the paper.
Please add figures and photos to the article.
A map with the location of the cities where the research was conducted is missing.
Please add photos of the places where the research was conducted.
Please describe in detail what elements of architecture, environment, culture were considered.
There is lack of information about the form which was filled in by the people who took part in the research.
Please also describe in graph form The Potential Restorative Scale contains 26 statements and measures the subject's experience of the environment's restor
Please change the order of the text in Chapter 2. First describe the equipment and parameters measured, then describe the procedure in detail.
Results should also be presented in graphs and not just tables.
The summary is too short and too general.
Please correct the entries in the literature list.